# Cyclic Behavior of L-Shaped RC Short-Limb Shear Walls with High-Strength Rebar and High-Strength Concrete

**Pinle Zhang** [1], **Jinyulin Wang** [1] **and Junfang Gao** [2,*]

1    Faculty of Civil Engineering and Mechanics, Kunming University of Science and Technology, Kunming 650500, China

2    Faculty of Mechanical and Electrical Engineering, Kunming University of Science and Technology, Kunming 650500, China

\*    Correspondence: gaojunfang789@163.com

**Abstract:** Six RC short-limb shear walls with an L-shaped section, constructed with high-strength rebar and high-strength concrete, were loaded to destruction with pseudo-static loading. Experimental results were discussed and compared with L-shaped RC short-limb shear walls with high-strength horizontal rebar in detail. Different failure modes were obtained, such as flexure-dominated failure for specimens with an aspect ratio of 2.8 and 2.15 and bending-shear failure for specimens with an aspect ratio of 1.75. With a decrease in the aspect ratio, ductility decreased, whereas with an increase in the axial compression ratio, the load-carrying capacity increased but ductility decreased accordingly. An obvious pinching effect was found in specimens with a smaller aspect ratio and a higher axial compression ratio. Using high-strength longitudinal rebar and high-strength concrete can obviously improve the lateral load-carrying capacity of walls; and using high-strength horizontal rebar can obviously improve the ultimate deformation capacity. The average ultimate drift ratios of HPLW and LW far exceeded the specification requirements of the Chinese GB50011-2010 code.

**Keywords:** high-strength rebar; high-strength concrete; short-limb shear wall; cyclic loading





## 1. Introduction

RC walls are usually used in tall buildings to withstand lateral loads for their effectiveness in limiting drifts [1]. RC short-limb shear walls refer to specific RC walls with the ratio of the cross-section height to width between 5 and 8 [2]. A short-limb shear wall structure has a flexible layout and good building function [3,4]. It has been very common to use short-limb shear walls with a rectangular section to optimize economy and design. Numerous studies have been conducted to investigate the seismic behavior of rectangular shear walls [5–8]. Previous experimental studies have shown that rectangular shear walls usually showed poor seismic performance [9], especially for rectangular short-limb shear walls [10]. For reasons of functionality, it is also a common practice to combine rectangular short-limb shear walls to form T-shaped or L-shaped short-limb shear walls [11].

Previous studies have been restricted to T-shaped [12–14], L-shaped [15–17], and rectangular walls [18,19] with ordinary-strength rebars and concrete. Very few documents have investigated the seismic performance of L-shaped short-limb shear walls with high-strength rebar and high-strength concrete. Previous experimental results also indicated that the web was the weakest part of conventional flanged walls, which generally failed with premature crushing in the free web boundary because of insufficient restraint [20]. In order to enhance the seismic performances of short-limb shear walls with an L-shaped section, we designed L-shaped short-limb shear walls with high-strength rebar and high-strength concrete. The high-strength longitudinal rebars serve to enhance the load-carrying capacity, especially when web is under tension. The high-strength horizontal rebars serve to confine concrete and postpone the bucking of longitudinal rebars, while increasing ductility

simultaneously. The high-strength concrete serves to improve the load-carrying capacity when the web is under compression.

In this paper, six L-shaped short-limb shear walls constructed with high-strength rebar and high-strength concrete (in short HPLW) were loaded to destruction under lateral cyclic loading. The effects of axial compression ratio, aspect ratio, and horizontal rebar spacing on the seismic performance of an HPLW were critically investigated. Experimental results are discussed and compared with an L-shaped short-limb shear wall with high-strength horizontal rebar (in short LW), and key issues related to seismic design are discussed in detail.

## 2. Experimental Program

### 2.1. Details of HPLW and LW Walls

Six L-shaped short-limb shear walls with high-strength rebar and high-strength concrete and six L-shaped short-limb shear walls with high-strength horizontal rebars were constructed and loaded with cyclic loading. The vertical height of all walls was 1400 mm. The design parameters of the walls are shown in Table 1, where the steel ratio, $\rho_s$, of the longitudinal rebars is defined as the ratio of the cross-sectional area of the longitudinal rebars to the total wall cross-sectional area; and the volumetric steel ratio, $\rho_v$, at the free web boundary is defined as the ratio of the volume of the stirrups to that of the wall [21]. The section parameters and rebar details of the walls are shown in Figure 1. The design strength grade of concrete used in the specimens of HPLW and LW was, respectively, C60 (nominal cubic compressive strength $f$cu,k = 60 MPa) and C40, respectively, and the average cubic compressive strength fcu of the concrete measured on cubes of 150 mm size was 62.3 and 47.2 MPa, respectively. Table 2 shows the mechanical properties of the rebars.

**Table 1.** Parameters of HPLW and LW.

| Specimens | Cross-Section Height-to-Width Ratio | Design Axial Load Ratio | Experimental Axial Load Ratio | $\rho_v$ | $\rho_s$ | Aspect Ratio |
|---|---|---|---|---|---|---|
| HPLW500-1 | 5.0 | 0.17 | 0.10 | 1.51% | 2.51% | 2.80 |
| HPLW500-2 | 5.0 | 0.50 | 0.30 | 3.02% | 2.51% | 2.80 |
| HPLW650-1 | 6.5 | 0.50 | 0.30 | 1.51% | 2.26% | 2.15 |
| HPLW650-2 | 6.5 | 0.50 | 0.30 | 3.02% | 2.26% | 2.15 |
| HPLW800-1 | 8.0 | 0.17 | 0.10 | 1.51% | 2.41% | 1.75 |
| HSLW800-2 | 8.0 | 0.34 | 0.20 | 3.02% | 2.41% | 1.75 |
| LW500-1 | 5.0 | 0.34 | 0.20 | 0.62% | 1.95% | 2.80 |
| LW500-2 | 5.0 | 0.50 | 0.30 | 1.25% | 1.95% | 2.80 |
| LW650-1 | 6.5 | 0.34 | 0.20 | 1.25% | 1.71% | 2.15 |
| LW650-2 | 6.5 | 0.17 | 0.10 | 1.25% | 1.71% | 2.15 |
| LW800-1 | 8.0 | 0.34 | 0.20 | 0.58% | 1.74% | 1.75 |
| LW800-2 | 8.0 | 0.17 | 0.10 | 1.16% | 1.74% | 1.75 |

**Table 2.** Mechanical properties of steel rebars of HPLW and LW walls.

| Steel Type | Yield Stress ($n$/mm$^2$) | Ultimate Stress ($n$/mm$^2$) | Elongation (%) | Elastic Modulus ($n$/mm$^2$) |
|---|---|---|---|---|
| D6 Rebar | 538 | 682 | 15 | 206,000 |
| D12 Rebar | 550 | 697 | 16 | 208,000 |
| #4 Rebar | 730 | 985 | 8 | 205,000 |
| #8 Rebar | 295 | 510 | 28 | 210,000 |
| #12 Rebar | 345 | 600 | 31 | 216,000 |

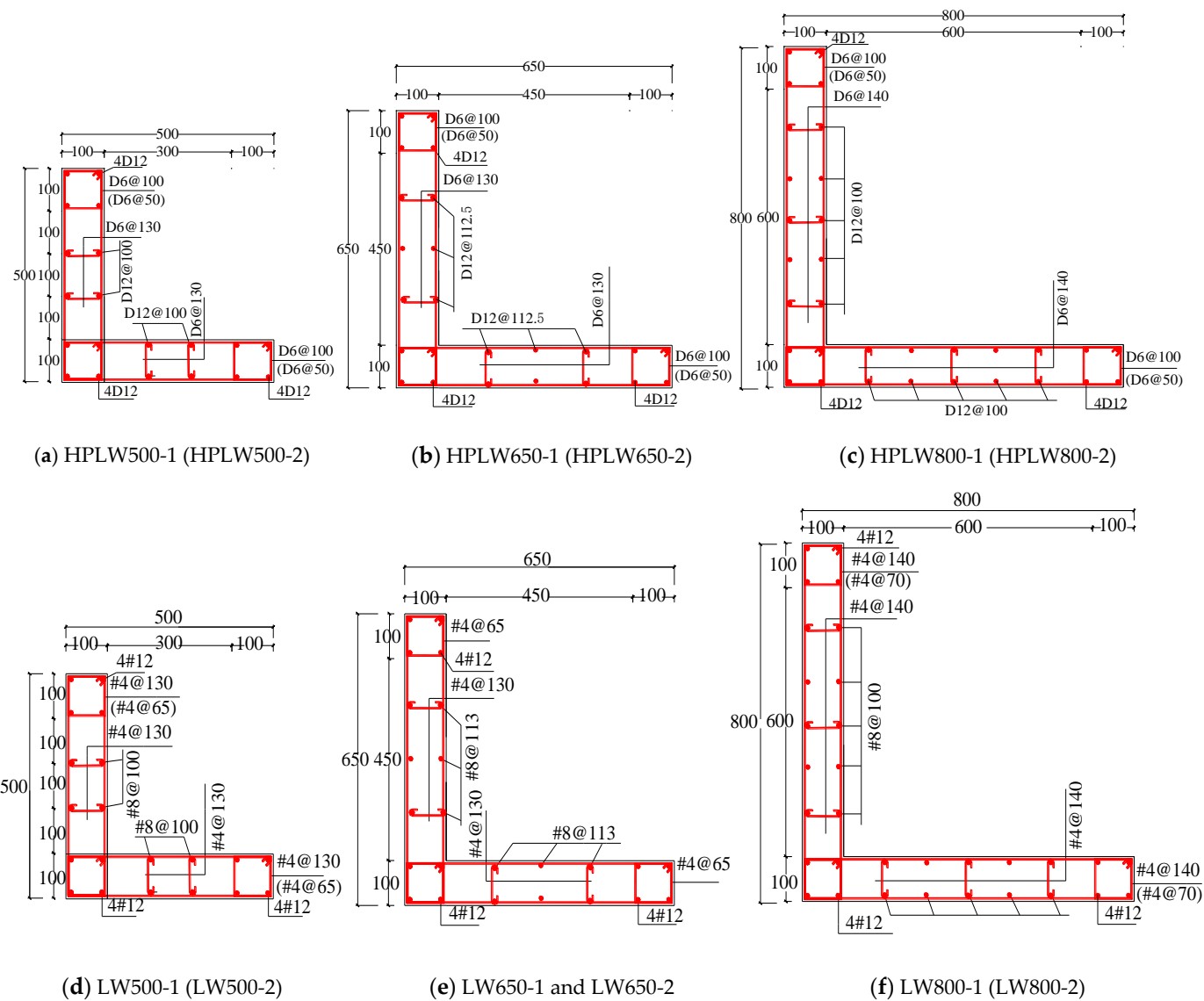

(**a**) HPLW500-1 (HPLW500-2)  (**b**) HPLW650-1 (HPLW650-2)  (**c**) HPLW800-1 (HPLW800-2)

(**d**) LW500-1 (LW500-2)  (**e**) LW650-1 and LW650-2  (**f**) LW800-1 (LW800-2)

**Figure 1.** Dimensions and reinforcement details of HPLW and LW walls.

### 2.2. Test Setup and Loading Sequence

The axial load was first applied to the center of the short-limb shear wall using the vertical jack and kept constant during the whole loading process. Figure 2 exhibits the test setup including loading devices. The lateral loading protocols adopted in the tests are illustrated in Figure 3. The loading history was started by applying two identical displacement cycles with increments of ±2 mm up to 10 mm, followed by increments of ±4 mm up to failure. Each test continued until the walls dropped to 85% of the maximum lateral load. Figure 4 defines the positive and negative loading directions. A number of linear variable differential transducers (LVDTs) were put on the specimens to measure displacements as shown in Figure 5. Figure 6 exhibits the arrangements of strain gauges of longitudinal and transverse rebars in the web. Figure 7 exhibits the arrangements of strain gauges of longitudinal rebars in the flange.

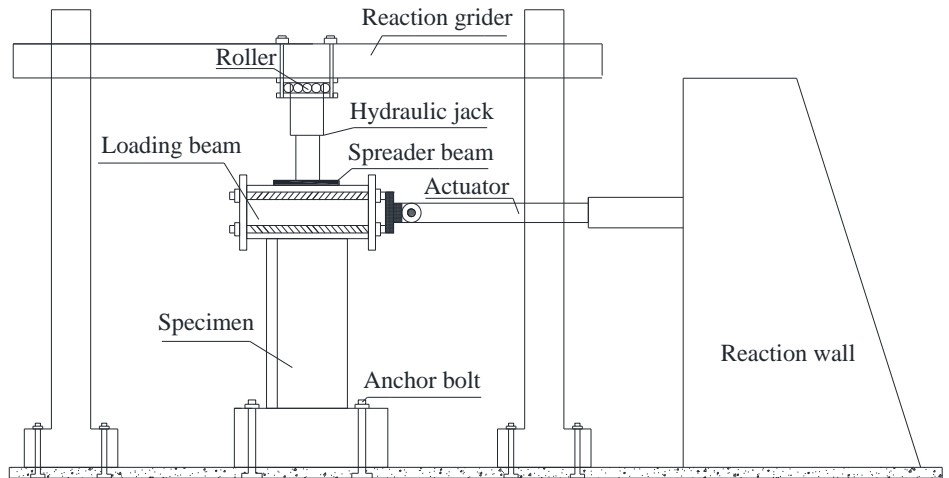

**Figure 2.** Test setup.

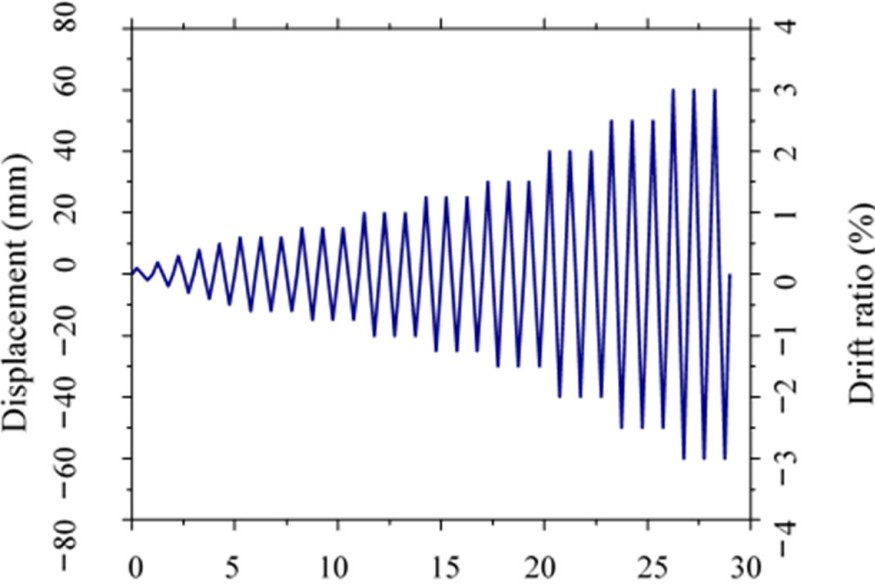

**Figure 3.** Displacement history of walls.

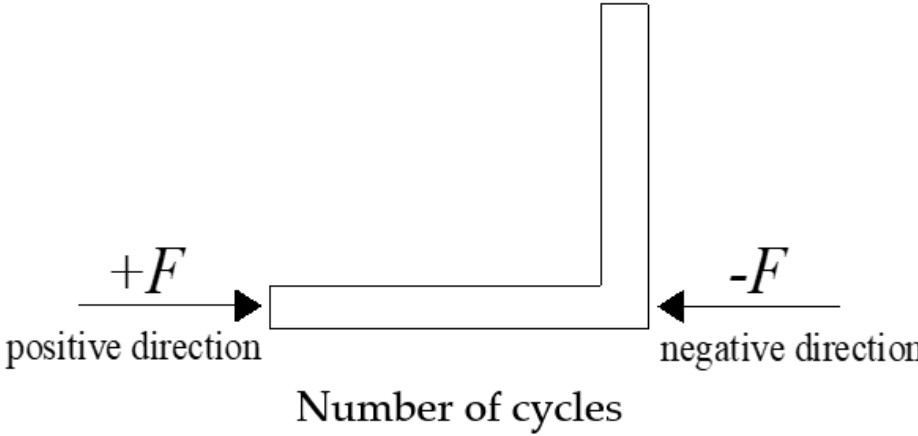

**Figure 4.** Loading directions.

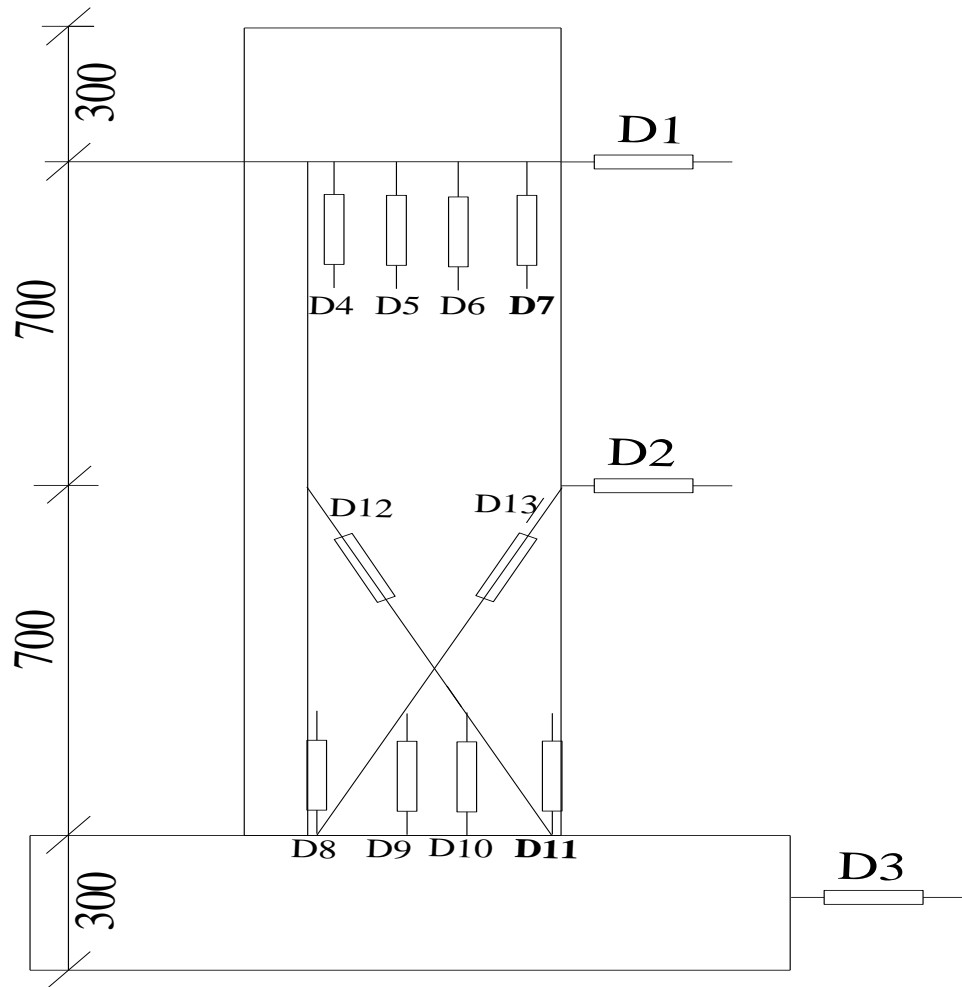

**Figure 5.** Arrangement of LVDTs in the web wall.

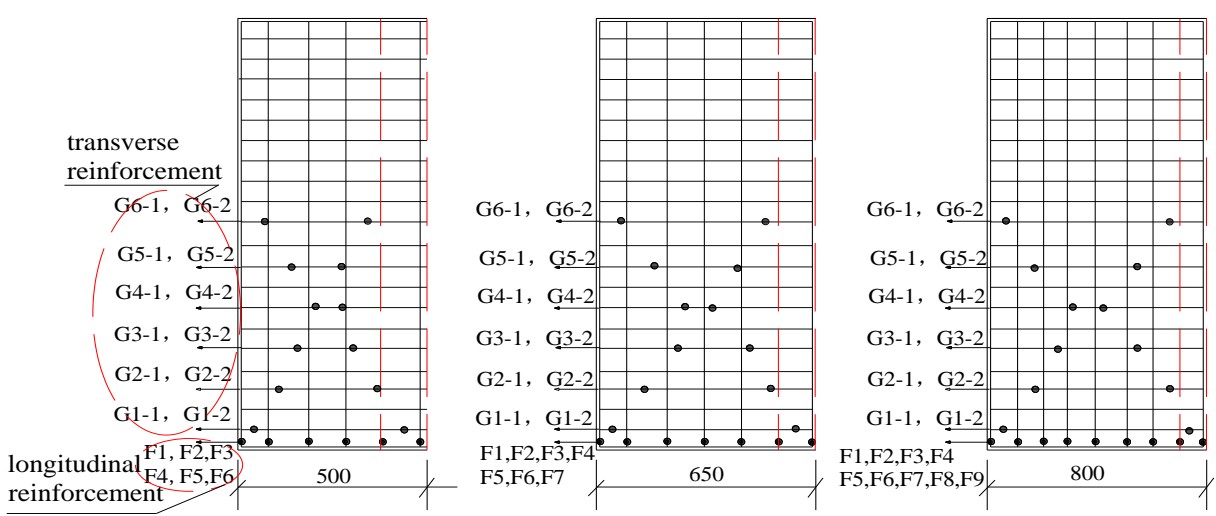

**Figure 6.** Arrangements of strain gauges of reinforcement in the web.

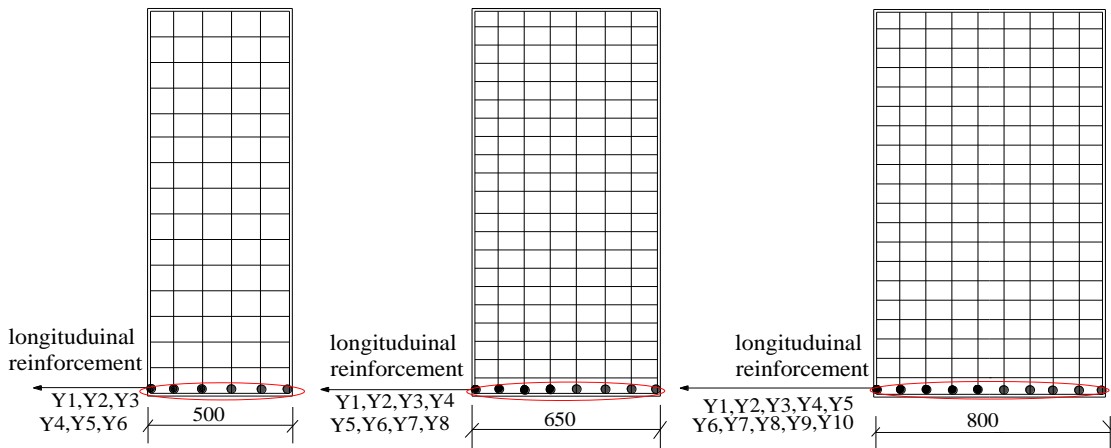

**Figure 7.** Arrangements of strain gauges of reinforcement in the flange.

## 3. Test Results and Discussion

### 3.1. Damaged Process and Mechanism

Different failure modes were observed, such as flexure-dominant failure mode for walls with a large aspect ratio of 2.8 and 2.15 bending-shear failure mode for walls with a small aspect ratio of 1.75. As similar behaviors of the other walls were observed, typical specimens of HPLW500-1 (flexure-dominant failure mode) and HPLW 800-1(bending-shear failure mode) were taken to illustrate the failure process and failure mechanism.

#### 3.1.1. Flexure-Dominant Failure

Figure 8 shows the crack patterns of HPLW500-1 at the failure stage. Several horizontal cracks appeared at the root of web at a drift ratio of 0.62% with a lateral load of 81.5 kN for positive loading direction. Shear cracks firstly formed at a drift ratio of 0.71%. The web boundary longitudinal reinforcing bars yielded at lateral loads of (+107.1 and −148.1 kN) at drift ratios of (0.80% and −1.00%). Peak lateral loads of 181.2 and −229.1 kN were reached at drift ratios of (3.11% and −2.43%). Extensive horizontal and shear cracks formed at the web, and modest spalling of the cover concrete occurred at the free web boundary at a drift ratio of 2.01%. During the drift ratio of −2.80%, cove concrete spalling at the wall boundaries extended up to approximately 200 mm from the wall–foundation interface. Wall lateral strength dropped to −170.8 kN or 74.6% of the peak strength under negative loading at a drift ratio of −3.0%, due to concrete crushing and buckling of boundary longitudinal reinforcements. However, minor concrete spalling and no vertical splitting were observed at the flange. Evidently, HPLW500-1 failed with an expected flexure-dominant behavior.

#### 3.1.2. Bending-Shear Failure

Figure 9 shows the crack patterns of HPLW800-1 at the failure stage. The crack pattern in this wall was very similar to that of the HPLW500-1; however, the inclined cracks formed to the top of the wall and the angle of inclined cracks was steeper than that in HPLW500-1. The first yielding of the boundary longitudinal reinforcement was observed at drift ratios of 0.86% and −1.14%. Peak lateral loads of 385.2 and 561.7 kN were reached at the same drift ratio of 2.70% for both positive and negative loading directions. At the end of testing, the wall panel was almost separated into two parts by the diagonal cracks. During the drift ratio of −2.91%, the bottom corners of concrete were crushed, and boundary longitudinal reinforcements buckled seriously under flexural compressive stresses. Obviously, HPLW800-1 failed with a combined shear and flexure mode.

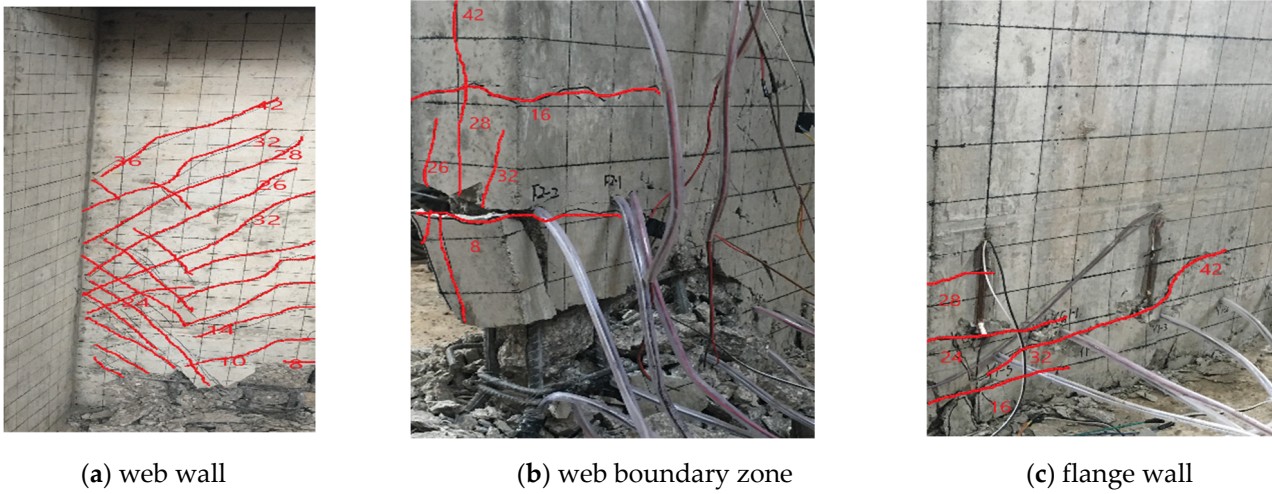

| (**a**) web wall | (**b**) web boundary zone | (**c**) flange wall |

**Figure 8.** Crack patterns of HPLW500-1 at failure stage.

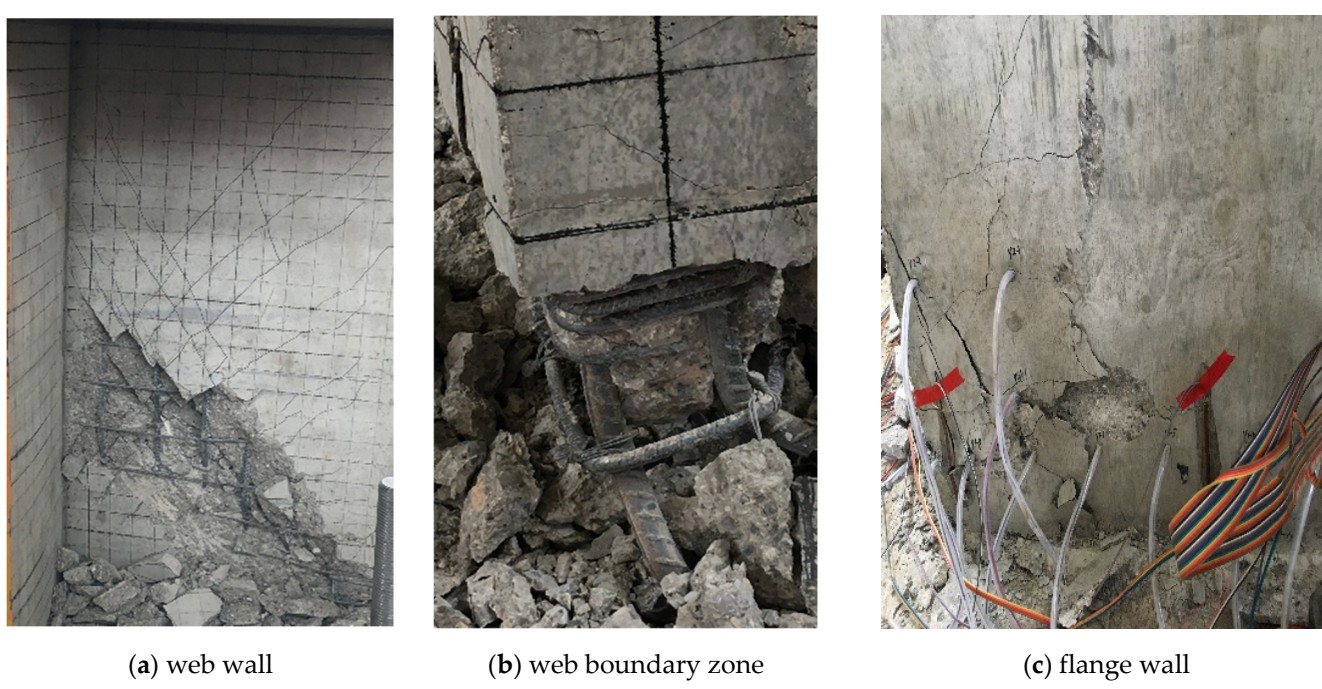

| (**a**) web wall | (**b**) web boundary zone | (**c**) flange wall |

**Figure 9.** Crack patterns of HPLW800-1 at failure stage.

*3.2. Hysteresis Curves*

The lateral load–top displacement hysteretic curves of the short-limb shear wall are exhibited in Figure 10. As shown in Figure 10, the hysteresis curves were asymmetrical; HPLW and LW all exhibited higher load-bearing capacity but lower ductility when the wall was subjected to loads in the negative direction. With a decrease in the aspect ratio, the ductility decreased, and with an increase in the axial compression ratio, the load-bearing capacity increased but ductility decreased accordingly. An obvious pinching effect was found in walls with a smaller aspect ratio and a higher axial compression ratio. The smaller the aspect ratio and the higher the axial load ratio used, the closer the transverse rebars and the longer confined boundary elements should be used in the free web boundary to improve its deformation capacity. Compared with LW walls, HPLW walls had a higher peak point, a larger ultimate displacement, and overall fatter hysteresis curves, showing that the load-carrying capacity, deformation capacity, and energy dissipation capacity of the latter were improved.

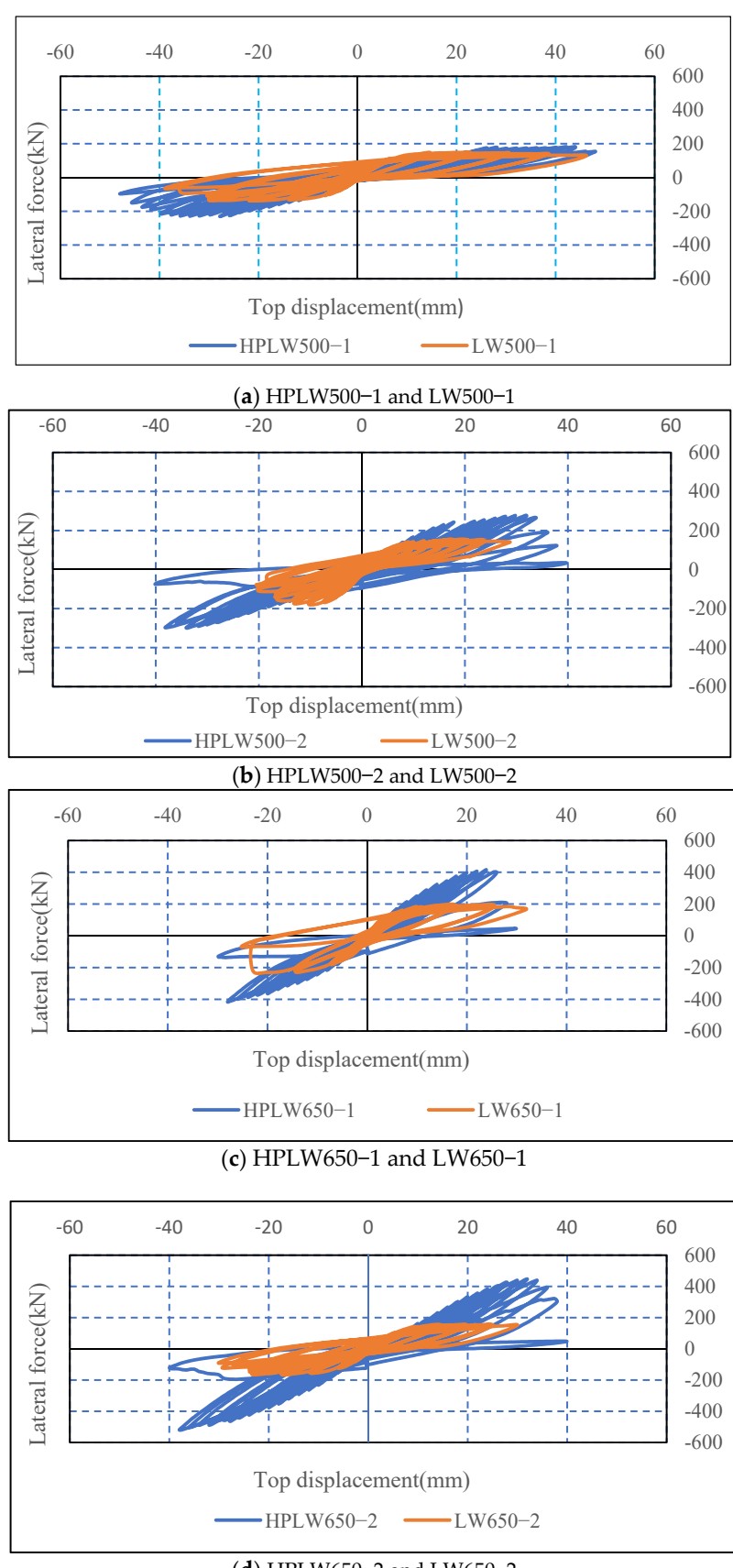

(**a**) HPLW500−1 and LW500−1

(**b**) HPLW500−2 and LW500−2

(**c**) HPLW650−1 and LW650−1

(**d**) HPLW650−2 and LW650−2

**Figure 10.** *Cont.*

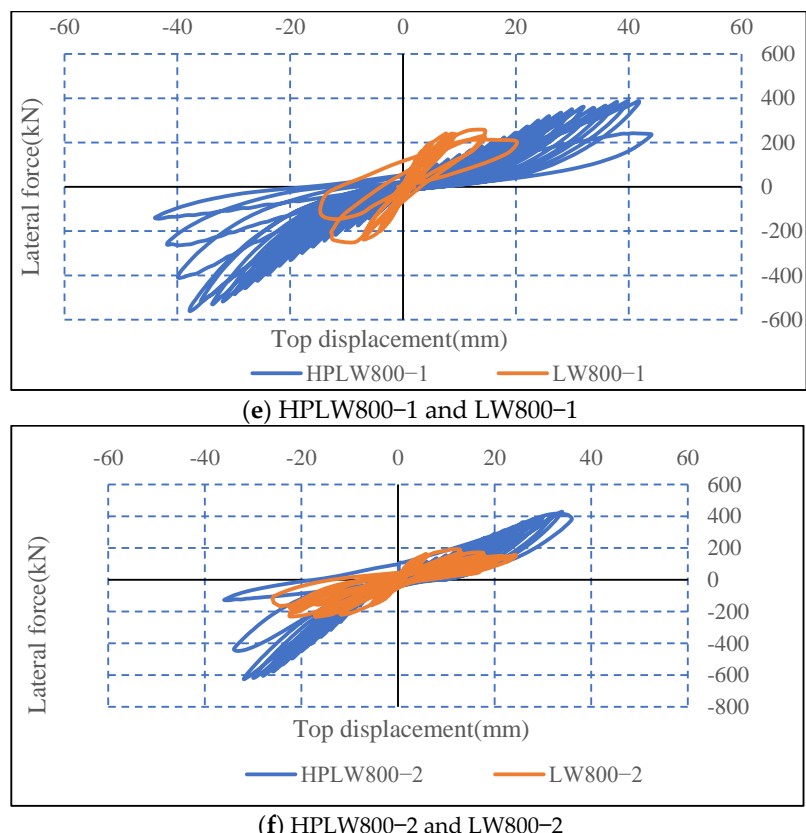

(**e**) HPLW800−1 and LW800−1

(**f**) HPLW800−2 and LW800−2

**Figure 10.** Lateral load–top displacement hysteresis curves of HPLW and LW wall.

### 3.3. Load-Bearing Capacity and Ductility

Table 3 exhibits the load-bearing capacity and ductility of HPLW and LW walls. Figure 11 exhibits the skeleton curves of HPLW and LW walls. As we can see in Table 3 and Figure 11, using high-strength longitudinal rebars and high-strength concrete obviously enhanced the load-bearing capacity. Compared with LW walls, the average yield lateral force and maximum lateral force of the HPLW specimens increased by 58.5% and 107.9%, respectively. The use of a high-strength stirrup also obviously improved the ultimate deformation capacity. All walls exhibited good deformation capacity. The average ultimate drift ratios of the HPLW and LW walls were, respectively, 1/37.6 and 1/68.0, which greatly exceeded the allowable interstory drift ratio value (1/120) of RC shear walls according to the design provisions of the Chinese GB50011-2010 code [21].

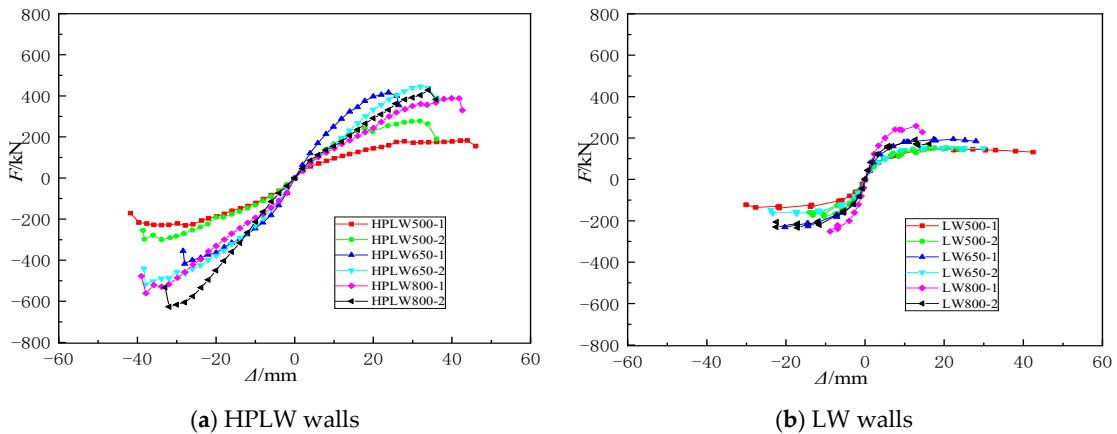

(**a**) HPLW walls

(**b**) LW walls

**Figure 11.** Skeleton curves of HPLW and LW walls.

**Table 3.** Load-bearing capacity and displacement ductility of HPLW and LW.

| Specimen | Loading Direction | $\Delta_y$ (mm) | $F_y$ (KN) | $\Delta_m$ (mm) | $F_m$ (KN) | $\Delta_u$ (mm) | $F_u$ (KN) | $\mu$ | $\theta_u$ |
|---|---|---|---|---|---|---|---|---|---|
| HPLW500-1 | + | 12.1 | 107.1 | 43.5 | 181.2 | 46.2 | 154.5 | 3.8 | 1/30.3 |
|  | − | 14.2 | 148.1 | 34.0 | 229.1 | 42.0 | 170.8 | 3.0 | 1/33.3 |
| HPLW500-2 | + | 9.9 | 168.8 | 33.8 | 300.1 | 38.1 | 241.1 | 3.9 | 1/36.8 |
|  | − | 12.0 | 188.6 | 32.1 | 279.0 | 34.3 | 265.1 | 2.9 | 1/40.8 |
| HPLW650-1 | + | 10.0 | 245.3 | 27.6 | 414.8 | 30.0 | 351.0 | 3.0 | 1/46.7 |
|  | − | 11.9 | 287.3 | 24.0 | 416.8 | 28.1 | 353.6 | 2.4 | 1/49.8 |
| HPLW650-2 | + | 12.0 | 264.4 | 35.9 | 522.6 | 38.0 | 396.9 | 3.2 | 1/36.8 |
|  | − | 14.0 | 267.1 | 31.3 | 440.5 | 35.8 | 394.5 | 2.6 | 1/39.1 |
| HPLW800-1 | + | 12.0 | 228.9 | 37.9 | 385.2 | 41.7 | 327.0 | 3.5 | 1/33.6 |
|  | − | 16.0 | 373.1 | 37.6 | 561.7 | 42.6 | 453.3 | 2.7 | 1/32.9 |
| HPLW800-2 | + | 12.0 | 256.2 | 33.8 | 427.9 | 36.0 | 394.1 | 3.0 | 1/38.9 |
|  | − | 12.0 | 268.2 | 31.8 | 626.2 | 33.5 | 449.6 | 2.8 | 1/41.8 |
| LW500-1 | + | 7.3 | 113.6 | 19.8 | 148.5 | 42.5 | 131.8 | 5.8 | 1/32.9 |
|  | − | 6.6 | 105.7 | 21.7 | 138.7 | 30.1 | 122.7 | 4.6 | 1/46.5 |
| LW500-2 | + | 4.9 | 101.8 | 20.5 | 152.2 | 25.0 | 144.6 | 5.1 | 1/56.0 |
|  | − | 3.8 | 121.4 | 10.3 | 178.3 | 14.1 | 158.7 | 3.7 | 1/99.3 |
| LW650-1 | + | 7.1 | 160.5 | 22.3 | 194.5 | 28.1 | 184.2 | 4.0 | 1/49.8 |
|  | − | 7.2 | 180.5 | 20.3 | 231.1 | 20.1 | 231.1 | 2.8 | 1/69.7 |
| LW650-2 | + | 5.1 | 100.0 | 22.0 | 153.2 | 29.9 | 150.2 | 5.9 | 1/46.8 |
|  | − | 5.7 | 123.1 | 17.3 | 169.8 | 23.8 | 148.2 | 4.2 | 1/58.8 |
| LW800-1 | + | 7.6 | 220.9 | 12.9 | 258.1 | 14.5 | 228.8 | 1.9 | 1/96.6 |
|  | − | 6.8 | 227.6 | 8.9 | 251.3 | 12.6 | 215.5 | 1.9 | 1/111.1 |
| LW800-2 | + | 5.8 | 160.0 | 12.7 | 192.2 | 16.2 | 172.1 | 2.8 | 1/86.4 |
|  | − | 5.5 | 153.2 | 17.1 | 233.8 | 22.4 | 206.0 | 4.1 | 1/62.5 |

Note: $F_y$ = yield lateral force; $\Delta_y$ = yield lateral displacement; $F_m$ = maximum lateral force; $\Delta_m$ = lateral displacement under the maximum lateral force; $F_u$ = ultimate lateral force; $\Delta_u$ = ultimate top displacement; $\mu$ = displacement ductility coefficient; $\theta_u$ = ultimate drift ratio. Here, '+' represents the loading direction in which the web is under compression; '−' represents the loading direction in which the web is under tension.

### 3.4. Energy Dissipation Capacity

The energy dissipation capacity in each cycle is evaluated by the equivalent viscous damping coefficient $h_e$, which can be calculated using Equation (1). Figure 12 explains the definition of the equivalent viscous damping coefficient. Figure 13 exhibits the equivalent viscous damping coefficient of HPLW specimens. It can be concluded that the energy dissipating capacity increases as the web confinement increases. The equivalent viscous damping coefficient under the failure stage was 0.15, 0.18, 0.20, 0.21, 0.14, and 0.18 for the specimens HPLW500-1, HPLW500-2, HPLW650-1, HPLW650-2, HPLW800-1, and HPLW800-2, respectively. The equivalent viscous damping coefficient under the failure stage was 0.16, 0.18, 0.23, 0.21, 0.21, and 0.12 for the specimens LW500-1, LW500-2, LW650-1, LW650-2, LW800-1, and LW800-2, respectively. Therefore, it can be concluded that the HPLW and LW specimens all have good energy dissipation capacity by using the high-strength stirrup.

$$h_e = \frac{1}{2\pi} \bullet \frac{S_{(ABC+CDA)}}{S_{\Delta OBE} + S_{\Delta ODF}} \tag{1}$$

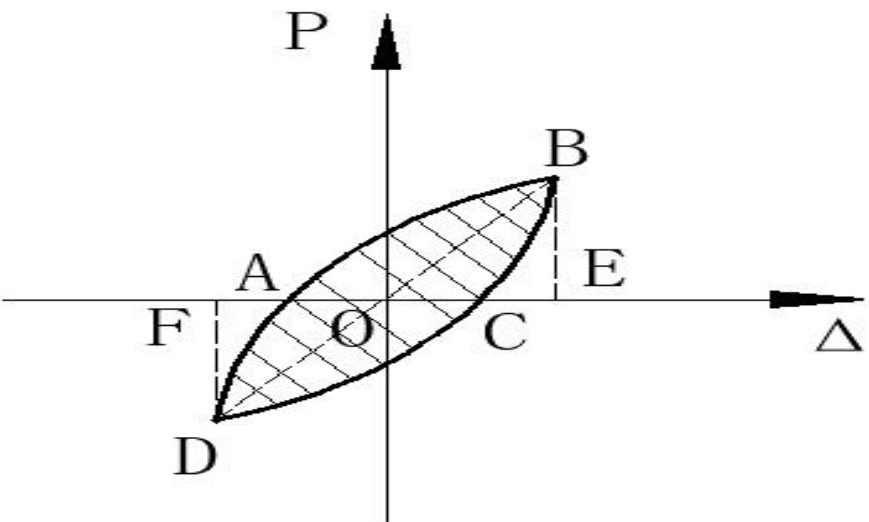

**Figure 12.** Calculation of equivalent viscous damping.

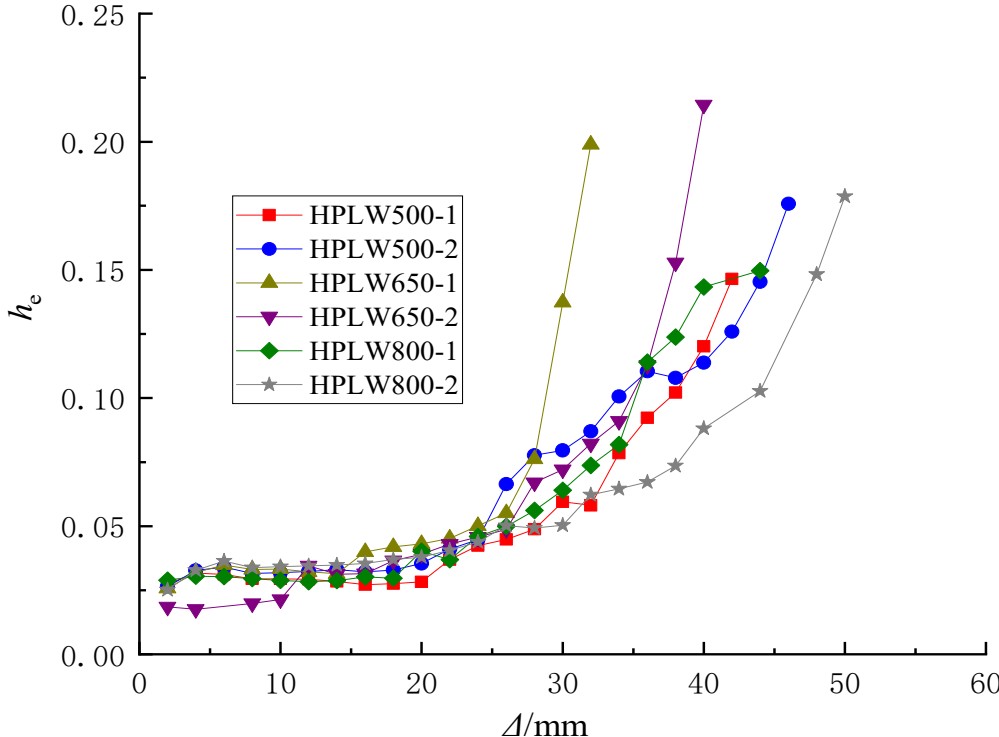

**Figure 13.** Equivalent viscous damping coefficient of viscous damping.

### 3.5. Strain Distribution of Vertical and Horizontal Rebars

Figure 14 exhibits the strain distribution of vertical rebars along the base of the wall length of typical specimens of HPLW500-2 and HPLW800-1. Figure 15 exhibits the strain distribution of horizontal rebars along the wall height of typical specimens of HPLW500-2 and HPLW800-1.

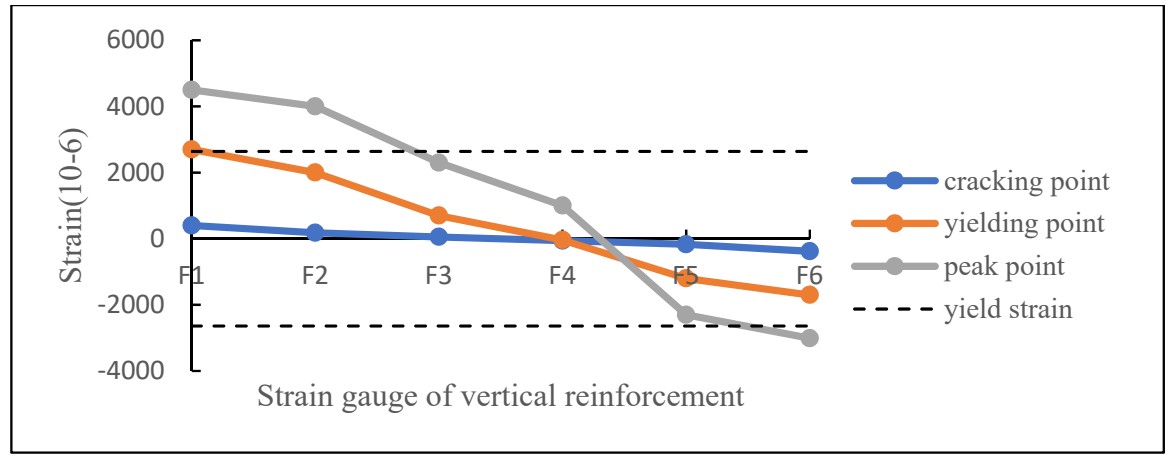

(**a**) HPLW500−2

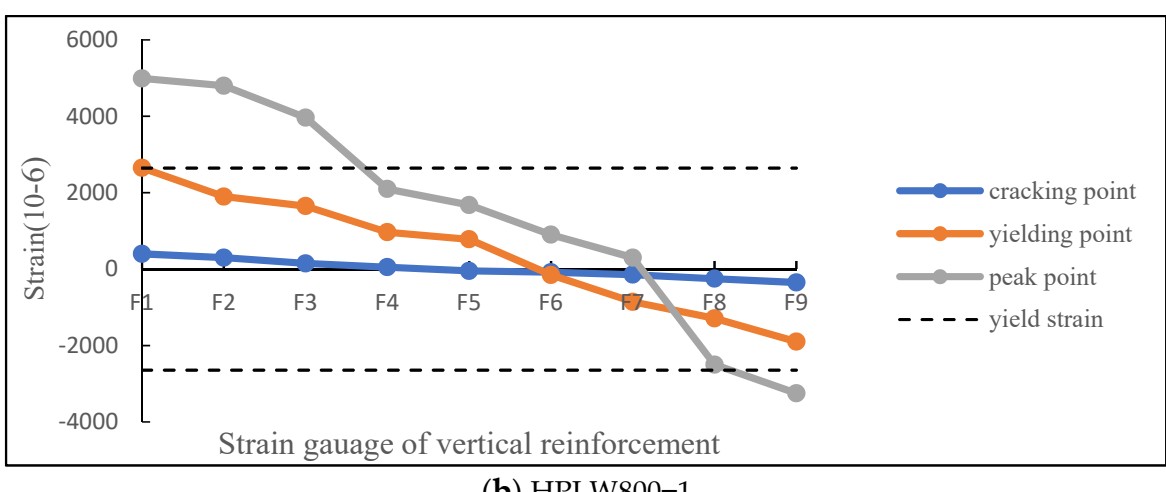

(**b**) HPLW800−1

**Figure 14.** Vertical reinforcement strain distributions along the wall length.

As we can see from Figure 14, the vertical reinforcement strains along the wall length were strictly linearly distributed at the yielding-point state, and the strains were approximately linearly distributed at the peak-point state. Therefore, the plane cross-section assumption may be used in the design of short-limb shear walls with an L-shaped section. As exhibited in Figure 15, most of the horizontal rebars were yielded. The yielded stirrups were mainly concentrated in the middle and lower parts of the web, indicating that the shear damage was mainly concentrated in the middle and lower part of the web, and the flange had little effect on improving the shear capacity. All high-strength stirrups at the free web boundary were yielded, indicating that the use of high-strength stirrups can strongly limit the lateral deformation of specimens and prevent premature failure of the web under compression.

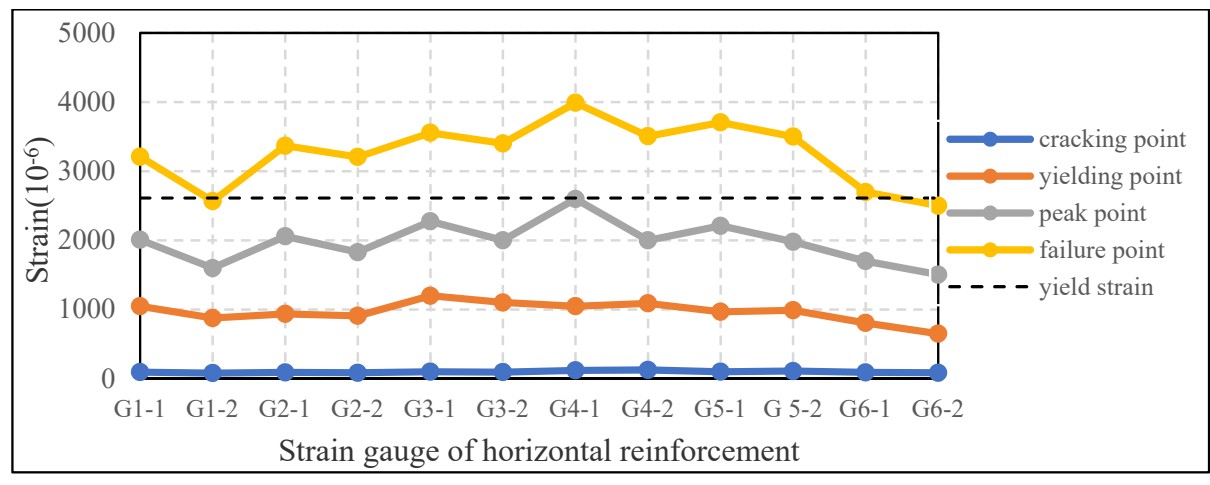

(**a**) HPLW500−2

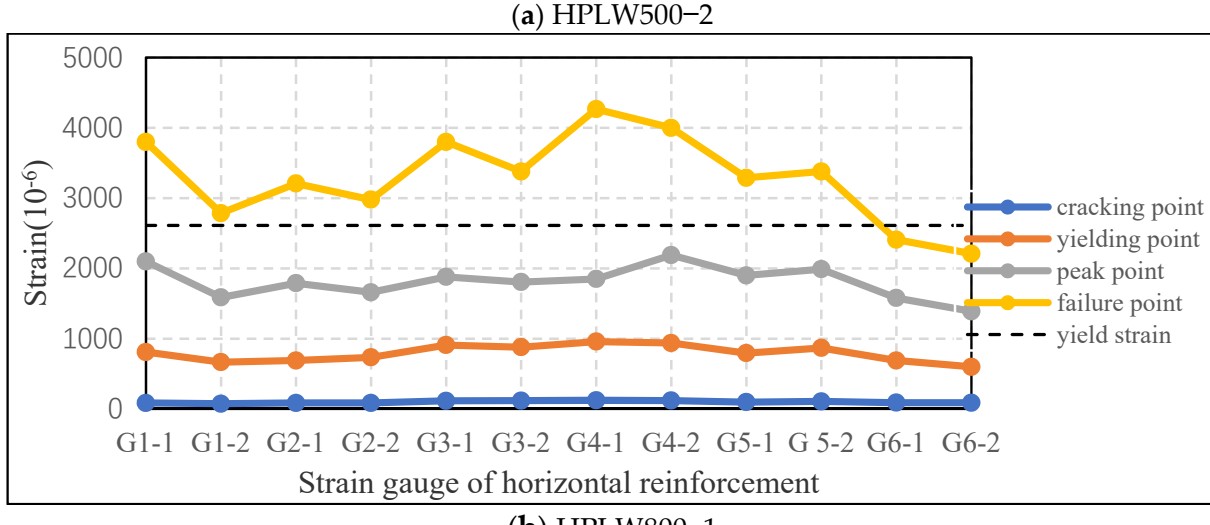

(**b**) HPLW800−1

**Figure 15.** Horizontal reinforcement strain distributions along the wall height.

## 4. Conclusions and Recommendations

In this paper, six L-shaped short-limb shear walls constructed with high-strength rebars and high-strength concrete and six L-shaped short-limb shear walls constructed with high-strength stirrups were loaded to destruction under cyclic loading. Based on the experimental results, the following conclusions can be obtained:

1.  Short-limb shear walls usually are dominated by the flexural effect. In this test, specimens with a larger aspect ratio exhibited the expected flexural-dominant behavior with concrete crushing as well as buckling or tensile failure of the longitudinal rebars at the free web boundary elements, such as specimens with an aspect ratio of 2.15 and 2.80. Specimens with a smaller aspect ratio tended to damage in bending-shear failure, such as specimens with an aspect ratio of 1.75.

2.  The free web boundary was still the weakest part of HPLW walls. Therefore, closer-spaced transverse rebars and longer confined boundary elements should be used at the free web boundary to prevent premature failure of the web in compression.

3.  With a decrease in the aspect ratio, the ductility decreased, and with an increase in the axial compression ratio, the load-bearing capacity increased but ductility decreased accordingly. An obvious pinching effect was found in specimens with a smaller aspect ratio and a higher axial compression ratio.

4.  The most effective method to increase the load-bearing capacity of the L-shaped short-limb shear wall is using high-strength longitudinal rebars or increasing the ratio of

longitudinal rebars at the free web boundary. The most effective method to improve the deformation capacity of the short-limb shear wall with L-shaped section is using high-strength transverse rebars or increasing the volumetric transverse reinforcement ratio at the free web boundary.

5.　All HPLW and LW walls showed excellent deformation capacity. The average ultimate drift ratios of the HPLW and LW walls were respectively 1/37.6 and 1/68.0, which greatly exceeded the allowable interstory drift ratio value (1/120) of RC shear walls according to the design provisions of the Chinese GB50011-2010 code.

6.　Compared with LW, HPLW walls had a higher peak point, a larger ultimate displacement, and overall fatter hysteresis curves, indicating that the load-carrying capacity, deformation capacity, and energy dissipation capacity of the latter were improved.

**Author Contributions:** Conceptualization, P.Z. and J.G.; validation, J.W.; formal analysis, P.Z. and J.G. All authors have read and agreed to the published version of the manuscript.

**Funding:** The research was funded by the National Natural Science Foundation of China (grant no.: 52168069, 51568028). The writers wish to express their sincere gratitude to the sponsor.

**Data Availability Statement:** Data available on request from the authors.

**Conflicts of Interest:** The authors declare no conflict of interest.

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
