# Peer review of "Cyclic Behavior of L-Shaped RC Short-Limb Shear Walls with High-Strength Rebar and High-Strength Concrete"

_applsci, doi:10.3390/app12168376_

Round 1
Reviewer 1 Report
Dear Autors,
I enjoyed reading this paper and found the results interesting. The introduction
is clear and frames the need for this research well. The literature review is
clear. The methods used in the paper are experimental methods. The methods
are appropriate for the work and clearly described. The analysis of the results
is sufficient. The authors report the outcomes and provide analysis of the
results. Although there is no separate discussion section the article has the
correct structure. The discussion was combined with the presentation of the
results and this is a good idea. English language and style are fine. Reference
to figures and tables has been done correctly. The figures and tables are clear.
I have not found any significant errors in the article and therefore recommend
its publication in its current form.
Author Response
Based on reviewer's comments, figures 10, 14, 15 have been modified like the other figures and to be clearer for further investigation numerically.
Reviewer 2 Report
I recommend accepting it. Only figures 10, 14, 15 need to me modified like the other figures and to be clearer for further investigation numerically.
1. What is the main question addressed by the research?
Is the high-strength longitudinal rebars serve to enhance the load-carrying capacity, especially when web is under tension; in addition to the high-strength horizontal rebars serve to confine concrete and postpone bucking of longitudinal rebars, increasing ductility simultaneously; The high-strength concrete serve to improve the load-carrying capacity when web is under compression.
2. Do you consider the topic original or relevant in the field? Does it address a specific gap in the field?
I think the parameters studied covered this gap which related to the use of high strength bars
The parameters were axial compression ratio, aspect ratio, horizontal rebar spacing on seismic performance of HPLW
3. What does it add to the subject area compared with other published material?
The above mentioned parameters
4. What specific improvements should the authors consider regarding the methodology? What further controls should be considered?
I think it was enough for the manuscript to include these experimental testing and results
5. Are the conclusions consistent with the evidence and arguments presented and do they address the main question posed?
yes
6. Are the references appropriate?
Yes
7. Please include any additional comments on the tables and figures.
As I said there are several of figures need to revisited and clarifying fig. 10, 14 and 15. The authors can modify them like the other figures; 1, 8, 9, 11
Author Response

(The authors gave the same response as above.)

Reviewer 3 Report
Experimental testing was performed of six RC short-limb shear walls with L-shaped section. Failure modes were discussed. The effects of high-strength concrete, high-strength longitudinal rebar and high-strength horizontal rebar were analyzed. The results obtained in the paper are useful in context of constructing of RC short-limb shear walls. I recommend the paper for publication.
Author Response

(The authors gave the same response as above.)
